# Neoadjuvant Chemoradiotherapy Followed by Esophagectomy with Three-Field Lymph Node Dissection for Thoracic Esophageal Squamous Cell Carcinoma Patients with Clinical Stage III and with Supraclavicular Lymph Node Metastasis

**DOI:** 10.3390/cancers13050983

**Published:** 2021-02-26

**Authors:** Yusuke Sato, Satoru Motoyama, Yuki Wada, Akiyuki Wakita, Yuta Kawakita, Yushi Nagaki, Kaori Terata, Kazuhiro Imai, Akira Anbai, Manabu Hashimoto, Yoshihiro Minamiya

**Affiliations:** 1Esophageal Surgery, Akita University Hospital, Akita 010-8543, Japan; motoyama@doc.med.akita-u.ac.jp (S.M.); wakita@gipc.akita-u.ac.jp (A.W.); yutakawakita@gipc.akita-u.ac.jp (Y.K.); nagaki@med.akita-u.ac.jp (Y.N.); minamiya@med.akita-u.ac.jp (Y.M.); 2Department of Thoracic Surgery, Akita University Graduate School of Medicine, Akita 010-8543, Japan; trt0605@gipc.akita-u.ac.jp (K.T.); karo@doc.med.akita-u.ac.jp (K.I.); 3Department of Comprehensive Cancer Control, Akita University Graduate School of Medicine, Akita 010-8543, Japan; 4Department of Radiology, Akita University Graduate School of Medicine, Akita 010-8543, Japan; ywada@med.akita-u.ac.jp (Y.W.); anbai@doc.med.akita-u.ac.jp (A.A.); hashimms@med.akita-u.ac.jp (M.H.)

**Keywords:** esophageal cancer, esophageal squamous cell carcinoma, prognosis, neoadjuvant treatment, chemoradiotherapy, NACRT, three-field, supraclavicular LN metastasis

## Abstract

**Simple Summary:**

This study aimed to clarify the efficacy of neoadjuvant chemoradiotherapy (NACRT) followed by esophagectomy with three-field lymph node (LN) dissection for clinical Stage III patients and for clinical Stage IVB patients with supraclavicular LN metastasis as the only distant metastatic factor. We observed that NACRT followed by esophagectomy with three-field lymph node dissection is feasible and offers the potential for long-term survival of these patients. It is also suggested that supraclavicular LNs should be treated as regional LNs at least in patients with upper and middle thoracic esophageal squamous cell carcinoma (ESCC).

**Abstract:**

Background: Neoadjuvant chemoradiotherapy (NACRT) followed by esophagectomy is now the standard treatment for patients with resectable advanced thoracic esophageal squamous cell carcinoma (ESCC) worldwide. However, the efficacy of NACRT followed by esophagectomy with three-field lymph node dissection for clinical Stage III patients and for clinical Stage IVB patients with supraclavicular LN metastasis has not yet been determined. Methods: Between 2008 and 2018, 94 ESCC patients diagnosed as clinical Stage III and 18 patients diagnosed as clinical Stage IVB with supraclavicular LN metastasis as the only distant metastatic factor were treated with NACRT followed by esophagectomy with extended lymph node dissection at Akita University Hospital. Long-term survival and the patterns of recurrence in these 112 patients were analyzed. Results: The median follow-up period of censored cases was 60 months. The five-year OS and DSS rates among the clinical Stage III patients were 57.6% and 66.6%, respectively. The five-year OS and DSS rates among the clinical Stage IVB patients were 41.3% and 51.6%, respectively. The most frequent recurrence pattern was distant metastasis (69.2%) in the Stage III patients and LN metastasis (75.0%) in the Stage IVB patients. Conclusion: NACRT followed by esophagectomy with three-field LN dissection is feasible and offers the potential for long-term survival of clinical Stage III ESCC patients and even clinical Stage IVB patients with supraclavicular LN metastasis as the only distant metastatic factor. At least in patients with upper and middle thoracic ESCC, treating supraclavicular LNs as regional LNs seems to be appropriate.

## 1. Introduction

Clinical trials carried out in several countries have demonstrated the efficacy of neoadjuvant treatments for patients with resectable advanced esophageal cancer [1,2,3,4,5]. Based on results from those trials, neoadjuvant chemoradiotherapy (NACRT) followed by esophagectomy is now the standard treatment for these patients worldwide. However, most of the patients in those trials were from European countries and had esophageal adenocarcinoma (EAC), often located in the lower esophagus, and were mainly treated with transhiatal esophagectomy or Ivor Lewis esophagectomy [3,4]. Moreover, the numbers of dissected lymph nodes (LNs) were only around 20 in these trials [3,4,5].

By contrast, in Asia, Africa, and Central and South America, the predominant pathological subtype is esophageal squamous cell carcinoma (ESCC) [6,7], which is often located more orally, in the upper or middle esophagus, and raises the possibility that LN metastasis could spread to cervical, mediastinal and abdominal LNs [8,9]. Several Japanese trials [10,11,12,13,14] have demonstrated the efficacy of extended cervical and upper mediastinal lymph node dissection for thoracic ESCC patients. Consequently, esophagectomy with extended lymphadenectomy, including the cervical and upper mediastinal LNs (so called ‘three-field LN dissection’), is now the standard surgical procedure for resectable thoracic ESCC patients in Japan. Because the standard surgical procedures are different, it is difficult to directly adapt evidence regarding neoadjuvant treatment strategies from regions where EAC predominates to those where ESCC predominates.

Based on results from the Japan Clinical Oncology Group (JCOG) 9204 [15] and JCOG 9907 [16] trials, neoadjuvant chemotherapy (NAC; cisplatin + 5-fluorouracil: CF) followed by esophagectomy with three-field LN dissection is now the standard treatment for these patients in Japan. At present, JCOG 1109 [17], comparing neoadjuvant CF, neoadjuvant DCF, and neoadjuvant chemoradiotherapy (NACRT, CF + 41.4 Gy/23 fractions of radiation) is ongoing, and we will need to wait a couple of years for evidence as to which neoadjuvant treatment is most appropriate before esophagectomy with three-field LN dissection. However, in 2008, prior to the start of JCOG 1109, we began employing NACRT before esophagectomy with three-field LN dissection for patients with resectable advanced ESCC. The aim of the present study is to clarify the long-term outcomes of 94 clinical Stage III patients treated with NACRT followed by esophagectomy with three-field LN dissection. We also aimed to clarify the long-term outcomes of 18 clinical Stage IVB patients who had supraclavicular LN metastasis as the only distant metastatic factor treated with NACRT followed by esophagectomy with three-field LN dissection. Results of the present study shows that this therapeutic strategy is feasible and provides long-term survival to patients with clinical Stage III ESCC. Moreover, although supraclavicular LN metastasis is defined as distant metastasis (M1) in the eighth edition of TNM classification of Malignant Tumors by the UICC [18], this treatment offers the potential for long-term survival, even in those patients.

## 2. Patients and Methods

### 2.1. Patients

This study was approved by the Ethics Committee of Akita University School of Medicine (#547) and all experiments were performed in accordance with the Helsinki Declaration. All study participants provided informed written consent. Among the 475 patients who received esophagectomy for esophageal cancer at Akita University Hospital between January 2008 and December 2018, 94 Stage III patients and 18 Stage IVB patients with supraclavicular LN metastasis as the only distant metastatic factor were retrospectively analyzed. The clinicopathological features of these patients are summarized in Table 1. They were all between 20 and 80 years old and had an Eastern Cooperative Oncology Group performance status (ECOG PS) of 0–1 before treatment. They were diagnosed as clinical Stage III or Stage IVB based on the eighth edition of the TNM classification of Malignant Tumors by the Union for International Cancer Control (UICC) [18]. Thoracic descending aorta dorsal LNs (112aoP in the 11th Edition of Japanese Classification of Esophageal Cancer) [8,9] were defined as regional LNs based on the eighth edition of the TNM classification of Malignant Tumors by the UICC [18], and patients with these LN metastases were included in this study. Although supraclavicular LN metastasis is defined as distant metastasis (M1) in the 8th edition of the TNM classification of Malignant Tumors by the UICC [18], the 11th Edition of the Japanese Classification of Esophageal Cancer [8,9] defines the supraclavicular LN as a reginal LN based on the results of the abovementioned Japanese trials [10,11,12,13], which show the efficacy of three-field LN dissection for thoracic ESCC patients. We therefore included Stage IVB patients with supraclavicular LN metastasis as the only distant metastatic factor in this study.

The clinical tumor stages of all patients were decided by a cancer board composed of radiologists, oncologists, gastroenterologists, and surgeons based on the results of blood tests, upper gastrointestinal endoscopy + US, CT, and [18F]-FDG-PET. The clinical diagnoses of supraclavicular LN metastasis were not confirmed by pathological examination before treatment in suspected cases.

### 2.2. Neoadjuvant Chemoradiotherapy

The regimen for NACRT was concurrent treatment with CF plus radiation. The drug dosages used for CF were identical to those used in the JCOG 9204 [6], JCOG 9907 [7], and JCOG 1109 trials [8]. Briefly, 80 mg/m^2^ cisplatin were administrated on day 1, and 800 mg/m^2^ 5-fluorouracil were administered as a continuous infusion from day1 to day 5. This protocol was then repeated with a 3- to 5-week interval in between. A representative radiation field in a patient with middle thoracic ESCC is shown in Figure 1. External body radiation was delivered with anterior and posterior opposite-beam interpolation using a 10 MV X-ray beam at 1.8–2.0 Gy/day for 5 days each week for a total dose of 41.4 Gy in 23 fractions. Radiation fields were limited to the primary esophageal lesion, with a 3-cm craniocaudal margin, and to clinically metastatic LNs; there was no elective nodal area radiation. If supraclavicular LNs were clinically negative, these LNs were outside of radiation field. All radiation plans were developed by certificated radiation oncologists using three-dimensional conformal radiotherapy planning based on simulation CT. Grading of adverse events with NACRT was according to the Common Terminology Criteria for Adverse Events (CTCAE) Version 5.0 (https://ctep.cancer.gov/protocolDevelopment/electronic_applications/ctc.htm#ctc_50, accessed on 8 January 2021).

### 2.3. Esophagectomy

Our standard operative procedure for thoracic ESCC patients is right thoracoscopic/robot-assisted or open esophagectomy with resection of the cardiac portion of the stomach. Also performed is three-field LN dissection of the upper to lower mediastinal (involving the periesophageal region: 105, 106, 108, 110, 111, 112 in the 11th Edition of Japanese Classification of Esophageal Cancer [8,9] and areas around the trachea and bilateral main bronchus: 107, 109), abdominal (involving the perigastric region:1, 2, 3 and areas around the celiac axis: 7, 8, 9, 11), and cervical (involving the bilateral periesophageal region: 101RL and supraclavicular region: 104RL) LNs. Cervical LN dissection was omitted for patients who meet following conditions; middle or lower esophageal main tumor, without objective evidence of upper mediastinal LN metastasis during thoracic procedure, with physical disadvantage (e.g., respiratory dysfunction, severe adverse event during NACRT).

Reconstruction typically involves insertion of a gastric conduit via the posterior mediastinal route [19,20,21]. Surgical complications are evaluated using the benchmark for complications and outcomes associated with esophagectomy [22]. Anastomotic leakage type I is defined as local defect requiring no change in therapy or treated medically or with dietary modification. Severe pneumonia is defined as respiratory failure requiring reintubation. Recurrent nerve palsy type I is defined as unilateral transient injury requiring no therapy based on the benchmark.

### 2.4. Pathological Response

The pathological response of the primary tumor was graded using the following response evaluation criteria for the effects of radiation, chemotherapy or both, published by the Japanese Esophageal Society [8,9]: Grade 0, no recognized cytological or histological therapeutic effect; Grade 1, slightly effective, with apparently viable cancer cells accounting for at least one-third of the tumor tissue; Grade 2, moderately effective with viable cancer cells accounting for less than one-third of the tumor tissue; and Grade 3, markedly effective, with no evidence of viable cancer cells (same as a complete response).

### 2.5. Statistical Analysis

For continuous variables, values are presented as the median (range, minimum-max). Length of survival was calculated from the date of the first neoadjuvant treatment to the patient’s death or the date of the last clinical follow-up. The Kaplan–Meier method was used to construct survival curves. The log-rank test was used to assess differences between the curves. Univariate analysis of prognostic factors was performed using a Cox proportional hazards model, and variables with *p* < 0.05 were included in the final multivariate model. All statistical analyses were performed using JMP Pro14 (Version 14.2.0, SAS Institute Inc., Cary, NC, USA).

## 3. Results

### 3.1. Neoadjuvant CRT Grade and Pathological Stage of Clinical Stage III and IVB Patients

Neoadjuvant CRT grade and pathological stage are summarized in Table 2. Among the 112 patients, 21 (18.8%) were diagnosed with pathological grade 3 (complete response), and 51 (45.5%) were pathological grade 2. Twenty-nine patients (25.9%) were diagnosed with ypT0, no residual main tumor, and 63 patients (56.2%) were diagnosed with ypN0. Thirty-nine patients (34.8%) were diagnosed with ypStage I, and 23 (20.5%) were ypStage II; thus, 55% patients showed down staging after neoadjuvant CRT. There were no significant differences between clinical Stage III and IVB with respect to grade or pathological T factor. Because Stage IVB patients were diagnosed with supraclavicular LN metastasis before treatment, ypN, ypM, and ypStage were significantly higher in these patients compared to Stage III patients.

### 3.2. Adverse Events and Reasons for Discontinuation during NACRT

Adverse events and reasons for discontinuation during NACRT are summarized in Table 3. The completion rate for neoadjuvant CRT was 86.6%. Frequent reasons of discontinuation were leukopenia (4.5%) and renal function deterioration (3.6%). There were no significant differences between clinical Stage III and IVB with respect to adverse events.

### 3.3. Esophagectomy with Three-Field LN Dissection after Neoadjuvant CRT

Details of the esophagectomy and its associated complications and prognoses are summarized in Table 4. Open esophagectomies were performed for 83 patients (74.1%), while thoracoscopic or robot-assisted esophagectomies were performed for 29 (25.9%) of the more recent patients. Cervical LN dissection was omitted for 8 patients only in clinical Stage III. The median surgical time was 575 min, and the median blood loss was 542.5 ml. The median number of all dissected LN was 49 (12–97). The median number of dissected LN in each area were as follows; cervical paraesophageal (101RL):3 (0–13), supraclavicular (104RL):14 (1–35), upper mediastinal (105,106,107,109):14 (3–50), lower mediastinal (108,110,111,112):5 (0–39) and abdominal (1,2,3,7,8,9,11):13 (0–45).

Median number of days between NACRT and esophagectomy was 40, and median length of the hospital stay after esophagectomy was 29 days. Anastomotic leakage (Type I or more) occurred in 16 patients (14.3%); 8 patients were Type I: local defect requiring no change in therapy, 3 patients were Type II: localized defect requiring interventional drain, 5 patients were Type III: localized defect requiring surgical therapy. Recurrent laryngeal nerve palsy (Type Ia or more) occurred in 26 patients (23.2%); 14 patients were Type Ia: unilateral transient injury requiring no therapy, 4 patients were Type Ib: bilateral transient injury requiring no therapy, 2 patients were Type IIa: unilateral injury requiring elective surgical procedure, 1 patient was Type IIb: bilateral injury requiring elective surgical procedure, 1 patient was Type IIIa: unilateral injury requiring acute surgical intervention, 4 patients were Type IIIb: bilateral Injury requiring acute surgical intervention. There were no significant differences between clinical Stage III and IVB patients with respect to factors associated with esophagectomy and complications. No deaths occurred within 30 days or 90 days after esophagectomy.

### 3.4. Five-Year Survival Analysis of Clinical Stage III and IVB Patients Treated with NACRT Followed by Esophagectomy with Three-Field LN Dissection

Kaplan–Meier analyses of OS and DSS among clinical Stage III and IVB patients are shown in Figure 2. The five-year OS rates among clinical Stage III and IVB patients were 57.6% and 41.3%, respectively (Figure 2A). There was no significant difference in five-year OS between clinical Stage III and IVB patients. The five-year DSS rates among clinical Stage III and IVB patients were 66.4% and 51.6%, respectively (Figure 2B). Similarly, there was no significant difference in five-year DSS between clinical Stage III and IVB patients.

### 3.5. Five-Year Survival Analysis of Pathologically Supraclavicular LN Metastasis-Positive and -Negative Patients

Kaplan–Meier analysis of OS and DSS among pathologically supraclavicular LN metastasis-negative and -positive patients are shown in Figure 3. Although all of these patients were diagnosed as clinical Stage III or more before treatment, the five-year OS rates of pathologically supraclavicular LN metastasis-negative and -positive patients were 56.8% and 23.8%, respectively (Figure 3A). However, the number of patients was small, and no significant difference in five-year OS between the two groups was detected. The five-year DSS rates among pathologically supraclavicular LN metastasis-negative and -positive patients were 66.6% and 23.8%, respectively (Figure 3B). Similarly, no significant difference in five-year DSS was detected between the two groups.

### 3.6. Pattern of Recurrence in 47 Patients after NACRT Followed by Esophagectomy with Three-Field LN Dissection

At the time of this analysis, 47/112 patients (42.0%) have experienced recurrence of their ESCC after NACRT followed by esophagectomy with extended LN dissection. The patterns of recurrence in these patients are summarized in Table 5. Distant metastasis (most frequently in a lung) occurred in 23/47 patients (48.9%), and dissemination occurred in 6/47 patients (12.8%). Non-regional LN metastasis (most frequently in an abdominal paraaortic LN) occurred in 10/47 patients (21.3%). Thus, 80% of recurrences were outside the surgical and radiation fields. Among the clinical Stage III patients, the most frequent recurrence pattern was distant metastasis (21/39, 53.8%). On the other hand, the most frequent recurrence pattern was LN metastasis (6/8, 75.0%) among the clinical Stage IVB patients.

## 4. Discussion

In the present study, we showed that NACRT followed by esophagectomy with three-field LN dissection provides long-term survival to patients with clinical Stage III ESCC. This treatment also offers the potential for long-term survival in clinical Stage IV ESCC patients with supraclavicular LN metastasis as the only distant metastatic factor. Among patients experiencing recurrence, the most frequent recurrence pattern in clinical Stage III patients was distant metastasis, while the most frequent pattern in clinical Stage IVB patients was LN metastasis.

In the JCOG 9907 trail [16], the five-year OS rate in the pre-CF group was 55%. However, that group was composed of 82 clinical Stage II patients and 82 clinical Stage III patients. Therefore, the 94 clinical Stage III patients in the present study treated with NACRT followed by esophagectomy with three-field LN dissection had a better five-year OS rate, 57.6%, than the combined Stage II and III patients in the JCOG 9907 trail.

The CROSS trial [4] reported that the five-year OS rate among ESCC patients treated with NACRT followed by esophagectomy was about 60%. However, that group included only 41 patients, and esophagectomy was performed using a transthoracic or transhiatal approach with two-field LN dissection. Moreover, the clinical staging of those 41 patients was not described.

The NEOCRTEC5010 trial [5] also reported that the five-year OS rate among 224 ESCC patients treated with NACRT followed by esophagectomy was about 60%. This group included 36 (16.1%) clinical Stage IIB patients and 188 (83.9%) clinical Stage III patients. McKeown or Ivor Lewis esophagectomy, including two-field lymphadenectomy with total mediastinal lymph node dissection, was performed, and a median of 20 (15 to 27) lymph nodes were dissected. By contrast, a median of 49 (12 to 97) lymph nodes were dissected in the present study. In the NEOCRTEC5010 trial, the 30-day and 90-day mortality rates were 0 and 0.5% (1/185), respectively, whereas both of 30-day and 90-day mortality rates were 0 in the present study.

Based on these results, we conclude that NACRT followed by esophagectomy with three-fields LN dissection is feasible and provides long-term survival to patients with clinical Stage III ESCC.

Clinical Stage IVB patients with supraclavicular LN metastasis as the only distant metastatic factor had five-year OS and DSS rates of 41.3% of and 51.6%, respectively, after NACRT followed by esophagectomy with three-field LN dissection. Although the number of patients in this group is small, there was no significant difference in five-year OS or DSS compared to clinical Stage III patients. This treatment strategy thus offers the potential for long-term survival, even in these clinical Stage IVB patients. In addition, since all 18 of these patients had an upper or middle thoracic tumor, it appears that treating supraclavicular LNs as regional LNs is appropriate, at least in patients with upper and middle thoracic ESCC.

Among the 47 patients who experienced a recurrence of their ESCC after NACRT followed by esophagectomy with three-field LN dissection, 80% of the recurrent sites were outside of the surgical and radiation fields. This means that NACRT followed by esophagectomy with extended lymphadenectomy enabled us to achieve powerful local control, but it was inadequate for control outside of the surgical and radiation fields. To achieve better control locally and systemically, a more powerful neoadjuvant chemotherapy regimen may be required. However, there have been very few trials directly comparing neoadjuvant chemotherapy with neoadjuvant chemoradiotherapy for ESCC patients. Two phase III randomized controlled trials, JCOG 1109 [17], comparing neoadjuvant CF, neoadjuvant DCF and NACRT (CF + 41.4 Gy/23 fractions of radiation), and HCHTOG1903 [23], comparing neoadjuvant paclitaxel + cisplatin and neoadjuvant paclitaxel + carboplatin + 41.4 Gy in 23 fractions, are ongoing. Results from these two randomized controlled trials should show us what is the most adequate neoadjuvant treatment for ESCC patients.

LN metastasis was the most frequent recurrence pattern in clinical Stage IVB patients with supraclavicular LN metastasis as the only distant metastatic factor. This is reasonable because these patients’ ESCCs are predisposed to favor lymphatic metastasis. We must therefore be alert to LN recurrence after esophagectomy in these patients. The mechanisms underlying lymphatic and hematogenous metastasis are completely different, which is another reason to treat supraclavicular LNs as regional LNs in ESCC patients.

We recently reported that prolonged interval between NACRT and esophagectomy had no impact on pCR rates or survival in the same cohort [24]. Chiu and colleagues reported same result [25]. On the other hand, van der Werf and colleagues reported that 13 weeks or longer interval was associated with a higher probability of having a pCR [26]. Therefore, this point is controversial.

Although the complete response rate after NACRT was only 18.8% in the present study, the rate reported by the CROSS trial was 48.6% [4]. Based on that high complete response rate, the necessity for planned esophagectomy has been questioned, especially for good responders to NACRT [27]. On the other hand, JCOG 1406-A [28], an exploratory analysis using pooled data from two prospective trials, JCOG9906 [29] and JCOG9907 [16], showed that overall survival was significantly better in the neoadjuvant CF + esophagectomy group, followed by the esophagectomy alone group, than in the definitive chemoradiotherapy group. Esophagectomy is thus an essential treatment for clinical Stage II/III ESCC patients. Recently, salvage esophagectomy in cases of residual disease after definitive chemoradiotherapy has drawn attention as a new treatment strategy for these patients [30,31,32].

Immune checkpoint inhibitors (ICIs) are now widely used in the treatment of ESCC patients [33]. ICIs are reportedly compatible with radiotherapy, and a number of clinical trials involving ICIs with radiotherapy for ESCC are ongoing [34]. Multimodal therapy centered on esophagectomy is required for treatment of clinical Stage II/III ESCC patients. A suitable combination of esophagectomy with chemotherapy, chemoradiotherapy, and ICIs may open up new possibilities for long-term survival of these patients.

The main limitations of the present study are its retrospective nature and the small number of clinical Stage IVB patients. Because the number of clinical Stage IVB patients was low, it is important that we continue to accumulate treatment results for these patients so as to produce more reliable survival curves. Another limitation is inaccuracy of clinical diagnosis of LN metastasis. Basically, we have to depend on CT and [18F]-FDG-PET about diagnosis of mediastinal and abdominal LN metastasis. However, these examinations have risk of false-positive or false-negative. More reliable examinations are necessary for more appropriate treatment decisions.

## 5. Conclusions

We observed that NACRT followed by esophagectomy with three-field LN dissection is feasible and provides long-term survival to clinical Stage III ESCC patients. Moreover, although supraclavicular LN metastasis is defined as distant metastasis, this treatment offers the potential for long-term survival, even in these patients. At least in patients with upper and middle thoracic ESCC, treating supraclavicular LNs as regional LNs seems to be appropriate.

## Figures and Tables

**Figure 1 cancers-13-00983-f001:**
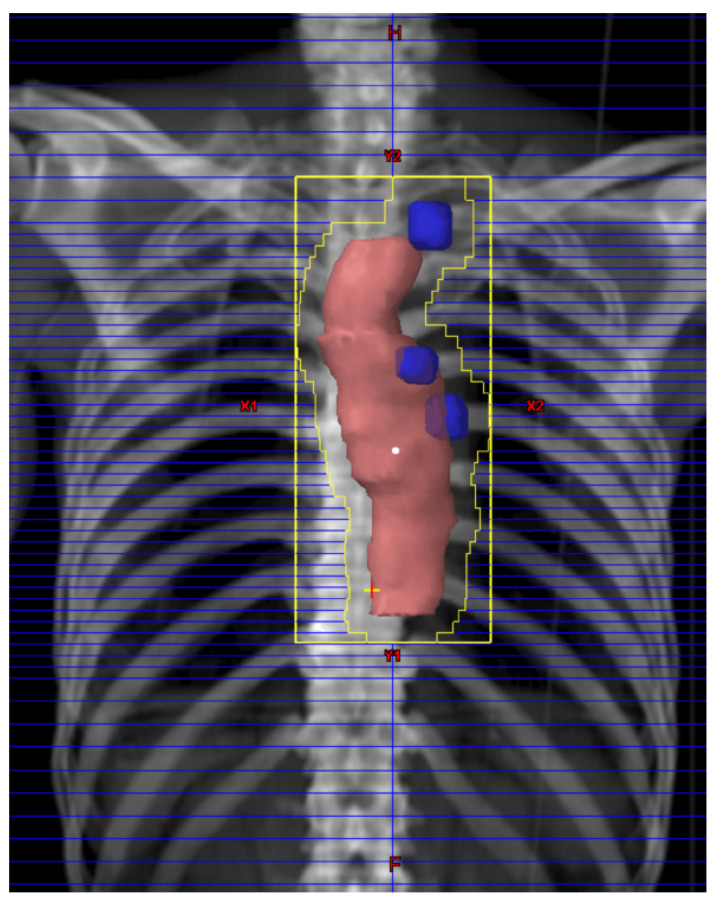
Representative radiation field in a middle thoracic ESCC patient. Shown is a beam’s eye view of the anterior field in a 69-year-old male with advanced middle thoracic ESCC staged as clinical T3N2M0. The pink structure is the primary esophageal lesion with a 3-cm craniocaudal margin. The blue structures are clinically metastatic lymph nodes.

**Figure 2 cancers-13-00983-f002:**
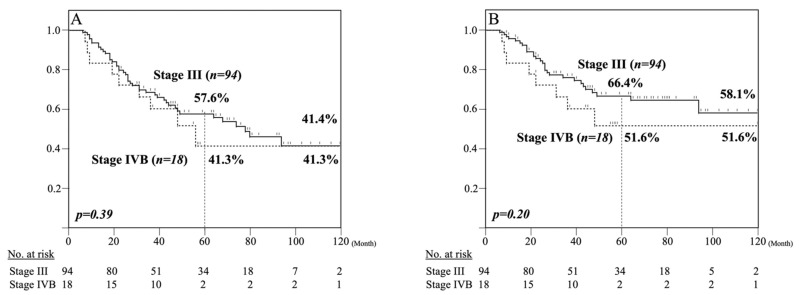
Kaplan–Meier analysis of clinical Stage III and IVB patients. The five-year OS was 57.6% among clinical Stage III patients and 41.3% among Stage IVB patients (**A**). The five-year DSS was 66.4% among clinical Stage III patients and 51.6% among Stage IVB patient (**B**).

**Figure 3 cancers-13-00983-f003:**
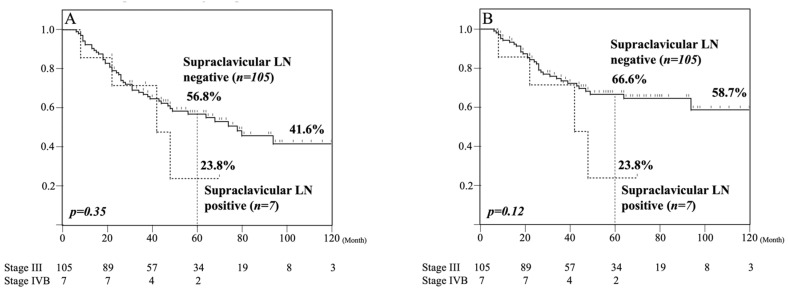
Kaplan–Meier analysis of pathologically supraclavicular LN metastasis-negative patients. The five-year OS was 56.8% and 23.8% among pathologically supraclavicular LN metastasis-negative and -positive patients, respectively (**A**). The five-year DSS was 66.6% and 23.8% among pathologically supraclavicular LN metastasis-negative and -positive patients, respectively (**B**).

**Table 1 cancers-13-00983-t001:** Clinicopathological features of all 112 ESCC patients.

Characteristics	All Patients (*n* = 112)	Clinical Stage III (*n* = 94)	Clinical Stage IVB (*n* = 18)	*p* Value
Sex				0.2479
Female	16 (14.3%)	15 (16.0%)	1 (5.6%)	
Male	96 (85.7%)	79 (85.7%)	17 (94.4%)	
Age at surgery	63.0	63.0	64.0	0.5895
	(41–77)	(43–75)	(41–77)	
Tumor location				0.0086
Upper	27 (24.1%)	22 (23.4%)	5 (27.8%)	
Middle	52 (46.4%)	39 (41.5%)	13 (41.5%)	
Lower	33 (29.5%)	33 (35.1%)	0	
Differentiation				0.3871
Well	25 (22.3%)	23 (24.5%)	2 (11.1%)	
Moderate	72 (64.3%)	58 (61.7%)	14 (77.8%)	
Poor	15 (13.4%)	13 (13.8%)	2 (11.1%)	
cT				0.0049
1	1 (0.9%)	0	1 (5.6%)	
2	1 (0.9%)	0	1 (5.6%)	
3	110 (98.2%)	94 (100%)	16 (88.8%)	
cN				0.0431
1	71 (63.4%)	62 (66.0%)	9 (50.0 %)	
2	40 (35.7%)	32 (34.0%)	8 (44.4%)	
3	1 (0.9%)	0	1 (5.6%)	
cM (supraclavicular LN metastasis)				<0.0001
Positive	18 (16.1%)	0	18 (100%)	
Negative	94 (83.9%)	94 (100%)	0	

**Table 2 cancers-13-00983-t002:** Clinicopathological features of all 112 ESCC patients after neoadjuvant CRT.

Characteristics	All Patients (*n* = 112)	Clinical Stage III (*n* = 94)	Clinical Stage IVB (*n* = 18)	*p* Value
Neoadjuvant treatment grade				0.7371
1	40 (35.7%)	35 (37.2%)	5 (27.8%)	
2	51 (45.5%)	42 (44.7%)	9 (50.0 %)	
3 (complete response)	21 (18.8%)	17 (18.1%)	4 (22.2%)	
ypT				0.3414
0	29 (25.9%)	23 (24.5%)	6 (33.2%)	
1	17 (15.2%)	12 (12.8%)	5 (27.8%)	
2	15 (13.4%)	14 (14.9%)	1 (5.6%)	
3	46 (41.0%)	41 (43.6%)	5 (27.8%)	
4a	3 (2.7%)	2 (2.1%)	1 (5.6%)	
4b	2 (1.8%)	2 (2.1%)	0	
ypN				0.0132
0	63 (56.2%)	54 (57.4%)	9 (50.0%)	
1	31 (27.7%)	26 (27.7%)	5 (27.8%)	
2	16 (14.3%)	14 (14.9%)	2 (11.1%)	
3	2 (1.8%)	0	2 (11.1%)	
ypM (supraclavicular LN metastasis)				<0.0001
positive	7 (6.3%)	2 (2.1%)	5 (27.8%)	
negative	105 (93.7%)	92 (97.9%)	13 (72.2%)	
ypStage				0.0039
I	39 (34.8%)	34 (36.2%)	5 (27.8%)	
II	23 (20.5%)	20 (21.3%)	3 (16.7%)	
IIIA	15 (13.4%)	13 (13.8%)	2 (11.1%)	
IIIB	25 (22.3%)	22 (23.4%)	3 (16.7%)	
IVA	3 (2.7%)	3 (3.2%)	0	
IVB	7 (6.3%)	2 (2.1%)	5 (27.8%)	

**Table 3 cancers-13-00983-t003:** Adverse events and reasons for discontinuation during NACRT.

Characteristic	All Patients (*n* = 112)	Clinical Stage III (*n* = 94)	Clinical Stage IVB (*n* = 18)	*p* Value
Leukopenia				0.7311
Grade 3	43 (38.4%)	38 (40.4%)	5 (27.8%)	
Grade 4	5 (4.5%)	4 (4.3%)	1 (5.6%)	
Neutropenia				0.5537
Grade 3	16 (14.3%)	13 (13.8%)	3 (16.7%)	
Grade 4	3 (2.7%)	2 (2.1%)	1 (5.6%)	
Anemia				0.6495
Grade 3	3 (2.7%)	2 (2.1%)	1 (5.6%)	
Grade 4	0	0	0	
Thrombopenia				0.1121
Grade 3	1 (0.9%)	1 (1.1%)	0	
Grade 4	3 (2.7%)	3 (3.2%)	0	
Hyponatremia				0.7753
Grade 3	8 (7.1%)	7 (7.5%)	1 (5.6%)	
Grade 4	0	0	0	
Neoadjuvant treatment completion				0.6562
Completed	97 (86.6%)	82 (87.2%)	15 (83.3)	
Not completed	15 (13.4%)	12 (12.8%)	3 (16.7%)	
Reason of discontinuation				-
Leukopenia	5 (4.5%)	4 (4.3%)	1 (5.6%)	
Renal function deterioration	4 (3.6%)	2 (2.1%)	2 (11.1%)	
Hyponatremia	2 (1.8%)	2 (2.1%)		
Thrombopenia	1 (0.9%)	1 (1.1%)		
Sepsis	1 (0.9%)	1 (1.1%)		
Osteomyelitis	1 (0.9%)	1 (1.1%)		
Rejection	1 (0.9%)	1 (1.1%)		

**Table 4 cancers-13-00983-t004:** Esophagectomy, complications, and prognoses for all 112 ESCC patients.

Characteristics	All Patients (*n* = 112)	Clinical Stage III (*n* = 94)	Clinical Stage IVB (*n* = 18)	*p* Value
LN dissection				0.1990
2-field	8 (7.1%)	8 (8.5%)	0	
3-field	104 (92.9%)	86 (91.5%)	18 (100%)	
Operative procedure				0.6980
Open	83 (74.1%)	69 (73.4%)	14 (77.8%)	
Thoracoscopic/robot-assisted	29 (25.9%)	25 (26.6%)	4 (22.2%)	
Organ for reconstruction				0.3816
Stomach	99 (88.4%)	82 (87.2%)	17 (94.4%)	
Colon	13 (11.6%)	12 (12.8%)	1 (5.6%)	
Reconstructive route				0.8281
Posterior mediastinal	97 (86.6%)	82 (87.2%)	15 (83.3%)	
Subcutaneous	15 (13.4%)	12 (12.8%)	3 (16.7%)	
Surgical time (min)				0.3724
	575	578	552	
	(386–928)	(386–928)	(407–704)	
Blood loss (mL)				0.5458
	542.5	550	535	
	(86–3366)	(86–3366)	(195–1833)	
Number of all dissected lymph nodes				0.8991
	49	49.5	49	
	(12–97)	(16–97)	(12–80)	
Cervical paraesophageal (101RL)				0.1498
	3	3	2	
	(0–13)	(0–13)	(0–6)	
Supraclavicular (104RL)				0.9389
	14	14	12.5	
	(1–35)	(1–35)	(3–32)	
Upper mediastinal (105, 106, 107, 109)				0.6829
	14	14	12.5	
	(3–50)	(3–50)	(4–44)	
Lower mediastinal (108, 110, 111, 112)				0.4759
	5	5	5	
	(0–39)	(0–39)	(0–13)	
Abdominal (1, 2, 3, 4, 7, 8, 9, 11)				0.1276
	13	13	9.5	
	(0–45)	(0–45)	(0–27)	
Days between neoadjuvant CRT and esophagectomy				0.8889
	40	40	39	
	(21–92)	(21–92)	(27–77)	
Days of hospital stay after esophagectomy				0.6059
	29	29	29	
	(15–168)	(15–168)	(16–111)	
Anastomotic leakage (Type I or more)	16 (14.3%)	14 (14.9%)	2 (11.1%)	0.6744
Respiratory failure requiring reintubation	6 (5.4%)	6 (6.4%)	0	0.2705
Recurrent laryngeal nerve palsy (Type Ia or more)	26 (23.2%)	22 (23.4%)	4 (22.2%)	0.9133
30-day mortality	0	0	0	
90-day mortality	0	0	0	
Recurrence of ESCC	47 (42.0%)	39 (41.5%)	8 (44.4%)	0.8160
Prognosis				0.3151
Alive	57 (50.9%)	47 (50.0%)	10 (55.6%)	
Alive after recurrence	8 (7.1%)	8 (8.5%)	0	
Deceased with ESCC	36 (32.2%)	28 (29.8%)	8 (44.4%)	
Deceased with other cancer	3 (2.7%)	3 (3.2%)	0	
Deceased with other diseases	8 (7.1%)	8 (8.5%)	0	

**Table 5 cancers-13-00983-t005:** Recurrence patterns in the 47 patients who experienced recurrence of ESCC.

Pattern of Recurrence	All Patients (*n* = 47/112, 42.0%)	Clinical Stage III (*n* = 39/94, 41.5%)	Clinical Stage IVB (*n* = 8/18, 44.4%)	*p* Value
distant metastasis	23 (48.9%)	21 (53.8%)	2 (25.0%)	0.1264
lung	12 (25.4%)	11 (28.2%)	1 (5.6%)	
liver	3 (6.4%)	3 (7.7%)		
kidney	3 (6.4%)	3 (7.7%)		
brain	2 (4.3%)	2 (5.1%)		
bone	2 (4.3%)	1 (2.6%)	1 (5.6%)	
skin	1 (2.1%)	1 (2.6%)		
dissemination	6 (12.8%)	6 (15.4%)		
non-regional LN	10 (21.3%)	7 (17.9%)	3 (37.5%)	
regional LN	7 (14.9%)	4 (10.3%)	3 (37.5%)	
intramural	1 (2.1%)	1 (2.6%)		

## Data Availability

The data presented in this study will be provided upon reasonable request.

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
