# Peer review of "Neoadjuvant Chemoradiotherapy Followed by Esophagectomy with Three-Field Lymph Node Dissection for Thoracic Esophageal Squamous Cell Carcinoma Patients with Clinical Stage III and with Supraclavicular Lymph Node Metastasis"

_cancers, 2021, doi:10.3390/cancers13050983_

Round 1

Reviewer 1 Report

Dear author,

It was a pleasure to review the paper “Neoadjuvant Chemoradiotherapy Followed by Esophagectomy 2 with 3-Field Lymph Node Dissection for Thoracic Esophageal 3 Squamous Cell Carcinoma Patients with Clinical Stage III and 4 with Supraclavicular Lymph Node Metastasis”, which I found of great interest. However, I have some remarks.

Firstly, the aim of the study is not clear. In particular, if the aim was to compare the two groups (stage III and stage IV ESCC) it should be specified and p values should be provided in every table.

In the section methods, it is not clear how the diagnosis of supraclavicular lymph node metastasis was obtained (if the staging was only clinical or confirmed by pathology). We suggest also to clarify if a supraclavicular lymph nodes dissection was performed in all cases of 3-field esophagectomy, even for distal esophageal cancer. If so, a comment in the section discussion should be provided because the high morbidity rate associated with such an extensive lymphadenectomy is only partially justified in case of a distal neoplasm.

We recommend to report the surgical complications according to the standardized method proposed by Low DE, Kuppusamy MK, Alderson D, et al. (Benchmarking Complications Associated with Esophagectomy. Ann Surg. 2019;269(2):291-298. doi:10.1097/SLA.0000000000002611).

I will be happy to review a revised version of the paper.

Sincerely

Author Response

Firstly, the aim of the study is not clear. In particular, if the aim was to compare the two groups (stage III and stage IV ESCC) it should be specified and p values should be provided in every table.

Response: As the reviewer suggested, we emphasized the aims of the study in Introduction (page 2, line 64) as follows, “The aim of the present study is to clarify the long-term outcomes of 94 clinical Stage III patients treated with NACRT followed by esophagectomy with 3-field LN dissection. We also aimed to clarify the long-term outcomes of 18 clinical Stage IVB patients who had supraclavicular LN metastasis as the only distant metastatic factor treated with NACRT followed by esophagectomy with 3-field LN dissection.”

We provided p values in every table to compare clinical Stage III and IVB patients. We also added following sentences in Results (page5, line152). “There were no significant differences between clinical Stage III and IVB with respect to grade or pathological T factor. Because Stage IVB patients were diagnosed with supraclavicular LN metastasis before treatment, ypN, ypM and ypStage were significantly higher in these patients compared to Stage III patients.

In the section methods, it is not clear how the diagnosis of supraclavicular lymph node metastasis was obtained (if the staging was only clinical or confirmed by pathology). We suggest also to clarify if a supraclavicular lymph nodes dissection was performed in all cases of 3-field esophagectomy, even for distal esophageal cancer. If so, a comment in the section discussion should be provided because the high morbidity rate associated with such an extensive lymphadenectomy is only partially justified in case of a distal neoplasm.

Response: As the reviewer suggested, we added following sentence in methods section (page 2, line 96). “The clinical diagnoses of supraclavicular LN metastasis were not confirmed by pathological examination before treatment in suspected cases.”

We added following sentences describing about omitting of cervical LN dissection in page5, line 123.

“Cervical LN dissection was omitted for patients who meet following conditions; middle or lower esophageal main tumor, without objective evidence of upper mediastinal LN metastasis during thoracic procedure, with physical disadvantage (e.g., respiratory dysfunction, severe adverse event during NACRT).”

In Results, page 6, line 166. “Cervical LN dissection was omitted for 8 patients only in clinical Stage III.”

We recommend to report the surgical complications according to the standardized method proposed by Low DE, Kuppusamy MK, Alderson D, et al. (Benchmarking Complications Associated with Esophagectomy. Ann Surg. 2019;269(2):291-298. doi:10.1097/SLA.0000000000002611).

Response: As the reviewer suggested, we cited recommended study instead of Clavien-Dindo classification and edited sentences page 5, line 128. “Surgical complications are evaluated using the benchmark for complications and outcomes associated with esophagectomy. (22). Anastomotic leakage type I is defined as Local defect requiring no change in therapy or treated medically or with dietary modification. Severe pneumonia is de-fined as respiratory failure requiring reintubation. Recurrent nerve palsy type I is defined as unilateral transient injury requiring no therapy (Dietary modification allowed) based on the benchmark.”

We very muchappreciate theconstructive suggestions from the Reviewer.

Reviewer 2 Report

1.The authors have identified based on their experience, that supraclavicular lymph node involvement for mid and upper ESCC, has a prognosis more similar to upper and mid esophageal ESCC with regional lymph node involvement as compared to those with distant metastatic disease.  This point should be emphasized.

2. It is interesting to note that approximately 56.2% of patients were ypN0 while ypT0 was 25.9%. As all patients were clinically staged with PET-CT, and all were N+, did the authors consider that NACRT rendered 56.2% of N+ patients to be N0 while only 25.9% were T0- is it possible, given the limitations of clinical staging that not all cN+ were actually histologically N+? The limitations of clinical staging are not recognized by some oncologists, leading to in appropriate treatment decisions. This point should also be emphasized.

3. Only clinical involved nodes were included in the radiation field, yet this would include all paraesophageal nodes based on the tumour location, especially once the margins are included. So in reality only supraclavicular nodes did not receive radiation if clinically negative. Please clarify.

4. The interval between NACRT varied from 21 to 92 days. Many authors have reported increased response rates including increased proportion of ypT0N0 with increasing interval between NACRT and surgery. It would be helpful for the authors to evaluate this in their cohort.

5. The lymph node harvest is impressive as is typical for Japanese studies. Western surgeons need to learn these techniques. However, the recurrent nerve palsy rate is significant and this has deterred Western surgeons. Please clarify if RLN palsies were uni or bilateral, temporary or permanent and whether or not tracheostomy was required.

6.It would be helpful for readers to know the distribution of the lymph nodes harvested, for example how many nodes were in the supraclavicular dissection, vs cervical, vs upper mediastinal, lower mediastinal and abdominal fields. These more general description of lymph node regions are easily understood by Western surgeons who are less familiar with the Japanese lymph node map.

7. Some surgeons have suggested that NACRT leads to lower lymph node harvest particularly if surgery is delayed as more radiation induced fibrosis occurs. Can the authors comment on this?

8. There are two very important points in this manuscript that warrant emphasis: 1)supraclavicular nodes are regional nodes for upper and mid third esophageal cancer ( and perhaps even lower third) 2) curative intent therapy is appropriate in the presence of supraclavicular nodal involvement. However,  the small number of patients with supraclav node involvement  limits these conclusions but suggests further study is indicated.

9. In patients without clinical supraclavicular node involvement, what was the proportion who had occult nodal metastases in this zone? Did they receive adjuvant RT to this area? How many of these patients had recurrence in this zone?

10. Can the authors offer any insight into the lower complete response rate with the NACRT regimen used in this study compared to the CROSS trial?  

11. It is interesting that the authors chose to report RLN and anastomotic leaks greater than CD1 but only greater than CD 3 for pneumonia. Reporting RLN palsy by CD1 or greater is reasonable given the bilateral cervical and supraclavicular lymph node dissection with the known greater risk of RLN injury. However why report CD grade 1-2 anastomotic leaks?

Author Response

1.The authors have identified based on their experience, that supraclavicular lymph node involvement for mid and upper ESCC, has a prognosis more similar to upper and mid esophageal ESCC with regional lymph node involvement as compared to those with distant metastatic disease.  This point should be emphasized.

Response: As the reviewer suggested, we added following sentence in Abstract (page 1, line 33) and Discussion (page 11, line 294). “At least in patients with upper and middle thoracic ESCC, treating supraclavicular LNs as regional LNs seems to be appropriate.”

  1. It is interesting to note that approximately 56.2% of patients were ypN0 while ypT0 was 25.9%. As all patients were clinically staged with PET-CT, and all were N+, did the authors consider that NACRT rendered 56.2% of N+ patients to be N0 while only 25.9% were T0- is it possible, given the limitations of clinical staging that not all cN+ were actually histologically N+? The limitations of clinical staging are not recognized by some oncologists, leading to in appropriate treatment decisions. This point should also be emphasized.

Response: As the reviewer suggested, we added following sentence in Discussion (page 11, line 290). “Another limitation is inaccuracy of clinical diagnosis of LN metastasis. Basically, we have to depend on CT and [18F]-FDG-PET about diagnosis of mediastinal and abdominal LN metastasis. However, these examinations have risk of false-positive or false-negative. More reliable examinations are necessary for more appropriate treatment decisions.”

  1. Only clinical involved nodes were included in the radiation field, yet this would include all paraesophageal nodes based on the tumour location, especially once the margins are included. So in reality only supraclavicular nodes did not receive radiation if clinically negative. Please clarify.

Response: We clarified about this point in Methods (page 3, line 110). “If supraclavicular LNs were clinically negative, these LNs were outside of radiation field.”

  1. The interval between NACRT varied from 21 to 92 days. Many authors have reported increased response rates including increased proportion of ypT0N0 with increasing interval between NACRT and surgery. It would be helpful for the authors to evaluate this in their cohort.

Response: We recently reported about this point in the same cohort. (Wakita A, et al. Verification of the Optimal Interval Before Esophagectomy After Preoperative Neoadjuvant Chemoradiotherapy for Locally Advanced Thoracic Esophageal Cancer. Ann Surg Oncol. 2020). In this paper, we showed that prolonged interval between NACRT and esophagectomy had no impact on pCR rates or survival. We described this point in Discussion (page 11, line 272).

  1. The lymph node harvest is impressive as is typical for Japanese studies. Western surgeons need to learn these techniques. However, the recurrent nerve palsy rate is significant and this has deterred Western surgeons. Please clarify if RLN palsies were uni or bilateral, temporary or permanent and whether or not tracheostomy was required.

Response: We reviewed patients who showed recurrent laryngeal nerve palsy and classified those patients based on the benchmark. We added following sentences in Results (page 7, line 178). “Recurrent laryngeal nerve palsy (Type Ia or more) occurred in 26 patients (23.2%); 14 patients were Type Ia: unilateral transient injury requiring no therapy, 4 patients were Type Ib: bilateral transient injury requiring no therapy, 2 patients were Type IIa: unilateral injury requiring elective surgical procedure, 1 patient was Type IIb: bilateral injury requiring elective surgical procedure, 1 patient was Type IIIa: unilateral injury requiring acute surgical intervention, 4 patients were Type IIIb: bilateral Injury requiring acute surgical intervention.”

  1. It would be helpful for readers to know the distribution of the lymph nodes harvested, for example how many nodes were in the supraclavicular dissection, vs cervical, vs upper mediastinal, lower mediastinal and abdominal fields. These more general description of lymph node regions are easily understood by Western surgeons who are less familiar with the Japanese lymph node map.

Response: As the reviewer suggested, we added these data in Table 4. We also added following description in Methods, “Also performed is 3-field LN dissection of the upper to lower mediastinal (involving the periesophageal region :105,106,108,110,111,112 in the 11th Edition of Japanese Classification of Esophageal Cancer (8,9) and areas around the trachea and bilateral main bronchus: 107,109), abdominal (involving the perigastric region:1,2,3 and areas around the celiac axis: 7,8,9,11), and cervical (involving the bilateral periesophageal region: 101RL and supraclavicular region: 104RL) LNs.

We also added following description in Results, “The median number of dissected LN in each area were as follows; cervical paraesophageal (101RL) : 3 (0-13), supraclavicular (104RL) : 14 (1-35), upper mediastinal (105,106,107,109) : 14 (3-50), lower mediastinal (108,110,111,112) : 5 (0-39) and abdominal (1,2,3,7,8,9,11) : 13 (0-45).”

  1. Some surgeons have suggested that NACRT leads to lower lymph node harvest particularly if surgery is delayed as more radiation induced fibrosis occurs. Can the authors comment on this?

Response: When we compared the number of harvested LN between NACRT group and NAC (neoadjuvant chemotherapy) group, which is different cohort, NACRT group showed significantly lower number compared to NAC group. That seems to be because of radiation induced fibrosis occurs. However, in the present cohort, there was no significant difference of the number of harvested LN between longer interval group and shorter interval group.

  1. There are two very important points in this manuscript that warrant emphasis: 1)supraclavicular nodes are regional nodes for upper and mid third esophageal cancer ( and perhaps even lower third) 2) curative intent therapy is appropriate in the presence of supraclavicular nodal involvement. However, the small number of patients with supraclav node involvement limits these conclusions but suggests further study is indicated.

Response: As the reviewer suggested, we emphasized this point in Discussion. (page 11, line 300).

  1. In patients without clinical supraclavicular node involvement, what was the proportion who had occult nodal metastases in this zone? Did they receive adjuvant RT to this area? How many of these patients had recurrence in this zone?

Response: In cStage III patients who were diagnosed with no supraclavicular LN metastasis, 2 patients (2.1%, 2/94) were pathologically diagnosed with supraclavicular LN metastasis. These patients did not receive any adjuvant RT. There was no patient who had recurrence in supraclavicular area in the present cohort.

  1. Can the authors offer any insight into the lower complete response rate with the NACRT regimen used in this study compared to the CROSS trial?

Response: In the CROSS trial, carboplatin + paclitaxel and 41.4 Gy of radiation were administrated. In the present study, we employed standard chemotherapy regimen in Japan, cisplatin + 5Fu and 41.4 Gy of radiation. We think the basic reason of the difference of complete response is patient cohort. There is not sufficient data about 41 ESCC patients who received NACRT and surgery in the CROSS trial, 84% of 178 patients (EAC+ESCC) were cT3 and 33% of patients were cN0. On the other hand, 98% of patients were cT3 and 100% were cN+ in our cohort.

  1. It is interesting that the authors chose to report RLN and anastomotic leaks greater than CD1 but only greater than CD 3 for pneumonia. Reporting RLN palsy by CD1 or greater is reasonable given the bilateral cervical and supraclavicular lymph node dissection with the known greater risk of RLN injury. However why report CD grade 1-2 anastomotic leaks?

Response: In our first submission, we used CD3b for pneumonia because CD3b: requiring reintubation is the clearest cutoff. CD1: only abnormality in blood test or chest Xray, CD2: bronchoscopy were less objective. As another reviewer suggested, we used the benchmark in revised version. Therefore, we used “Respiratory failure requiring reintubation” instead of CD3b. We also added detailed description about anastomotic leakage and RLN palsy in Results (page 7, line 180).

We very much appreciate the constructive suggestions from the Reviewer.